# Thermodynamic Behavior of Fe-Mn and Fe-Mn-Ag Powder Mixtures during Selective Laser Melting

**Jakob Kraner** [1,2,*], **Jožef Medved** [2], **Matjaž Godec** [1] and **Irena Paulin** [1]

1 Institute of Metals and Technology, Lepi pot 11, 1000 Ljubljana, Slovenia; matjaz.godec@imt.si (M.G.); irena.paulin@imt.si (I.P.)
2 Department of Materials and Metallurgy, Faculty of Natural Sciences and Engineering, University of Ljubljana, Aškerčeva cesta 12, 1000 Ljubljana, Slovenia; jozef.medved@ntf.uni-lj.si
* Correspondence: jakob.kraner@imt.si; Tel.: +386-41-865-166

**Abstract:** Additive manufacturing is a form of powder metallurgy, which means the properties of the initial metal powders (chemical composition, powder morphology and size) impact the final properties of the resulting parts. A complete characterization, including thermodynamic effects and the behavior of the metal powders at elevated temperatures, is crucial when planning the manufacturing process. The analysis of the Fe-Mn and Fe-Mn-Ag powder mixtures, made from pure elemental powders, shows a high susceptibility to sintering in the temperature interval from 700 to 1000 °C. Here, numerous changes to the manganese oxides and the αMn to βMn transformation occurred. The problems of mechanically mixed powders, when using selective laser melting, were highlighted by the low flowability, which led to a less controllable process, an uncontrolled arrangement of the powder and a large percentage of burnt manganese. All this was determined from the altered chemical compositions of the produced parts. The impact of the increased manganese content on the decreased probability of the transformation from γ-austenite to ε-martensite was confirmed. The ε-martensite in the microstructure increased the hardness of the material, but at the same time, its magnetic properties reduce the usefulness for medical applications. However, the produced parts had comparable elongations to human bone.

**Keywords:** FeMn alloys; powder metallurgy; differential scanning calorimetry; selective laser melting; ε-martensite

## 1. Introduction

Additive manufacturing (AM), a form of powder metallurgy, can create complex-shaped parts suitable for a wide range of applications. One of the most widespread industrial technologies used for AM is selective laser melting (SLM), which enables the processing of complex parts with appropriate microstructures. The SLM process can use the feedstock material from two or three elemental powders. Ti-35Nb [1] and Ti-6Al-4V [2] were successfully in situ created from elemental powders but resulted in a heterogeneous microstructure, which must be improved with post-processing heat treatment. Martinez et al. [3] researched Al-12Cu and concluded that the powder packing density must be maximized by varying particle sizes and shapes. Furthermore, the higher melting point temperature difference in elemental powders adversely effects the chemical composition homogeneity in in situ SLM-created alloys [3]. However, depending on the combination of elements, it is sometimes difficult to control the chemical composition of a part made using SLM. It is particularly problematic when SLM is applied to Zn [4] and Mg [5] alloys and other alloys with low-vapor pressure elements, such as alloys of iron (Fe) and manganese (Mn) [6]. This is unfortunate because FeMn alloys have good biocompatible and biodegradable properties for medical applications, especially if AM is used to produce the parts [7,8].

Tremendous progress in surgical techniques has been made by replacing permanent medical implants with temporary ones. Such implants offer the necessary support to the human body, but after the body recovers, the implants degrade without the need of any further invasive surgery and so improve the success rates for many surgical procedures. The Fe-based biodegradable group is currently dominated by pure-Fe and SS 316L [9,10]. However, a number of studies have recognized the potential of FeMn alloys [11,12], which corrode much quicker than pure Fe and SS 316L. For example, excellent corrosion rates were observed for alloys with 20 to 40 wt. % of Mn [13].

The best composition, as reported by Hermawan et al. [14], would be Fe30Mn. Using selective laser melting (SLM), they were able to achieve 520 MPa of the ultimate tensile strength (UTS), 240 MPa of the yield strength (YT) and 20% of the elongation (A). However, these are far from the properties of human bone [15]. Furthermore, the hardness of Fe30Mn, equal to 163 HV, as determined by Carluccio et al. [16], is not comparable to the hardness of human bone [17]. These high hardness values for Fe30Mn are a consequence of ε-martensite formation [18], resulting from rapid cooling rates that create highly non-equilibrium conditions. The ε-martensite transformation can be easily altered by changing the chemical composition. Liu et al. [19] were successful in stabilizing the γ-austenite and creating new biodegradable alloys with different additional elements. Chou et al. [5] stated that despite the presence of antiferromagnetic ε-martensite in Fe-Mn alloys, they could be used for craniofacial applications. For Fe-Mn alloys, corrosion tests were often performed, and the results compared with those achieved with pure Fe and SS 316L. The most interesting are the results of Hong et al. [20], where the addition of Mg improved the corrosion potential by 0.08 V compared to pure Fe, and Donik et al. [21], where the conventionally produced alloy was laser textured and the degradation rate increased from 62 to 863 nm/year. For a higher corrosion rate, a post-production treatment can be added to previously additive-manufactured samples. Wiesener et al. [22] created a material from the Fe–Mn–Ag ternary system by using pure, mechanically mixed powders and the SLM technique, despite the Fe and Ag establishing an isomorphous system. Similar alloys with a high Ag content were studied by Niendorf et al. [23] to explain the successful impact of microgalvanic corrosion, where the Ag phases are the local cathodes for accelerating the anodic corrosion. A small percentage (up to 1 wt. %) of Ag in Fe-Mn alloys could have a positive impact by increasing the rate of antiseptics.

The characterization of SLM samples made from mechanical mixtures of Fe, Mn and Ag powders was centered on thermodynamic aspects. The focus of our study was to determine whether it is possible to produce FeMn parts with appropriate properties using powder mixtures of the elements by the SLM technique. Despite the fact that the Fe–Ag phase diagram suggests that Fe and Ag do not mix in either the liquid or solid state, our focus was to prepare an Fe-Mn-Ag alloy by a rapid solidification process using SLM. Our investigation also focused on a thermodynamic understanding of high-vapor pressure Mn and its alloying impact. The differential scanning calorimetry (DSC) results indicate the influence of the chemical composition and especially the Ag addition on changes to the solidification starting temperatures and the solidification process itself. The multiple re-melting as a result of the DSC repetitions of heating, isotherm and cooling causes the evaporation of Mn that drastically changes the material´s chemical composition and creates a greater heterogeneity of the material than achieved with mechanically mixed powders. The ε-martensite transformation, $MnO_2$ melting between 700 and 900 °C and the susceptibility of the powder mixture to sintering were monitored and discussed as potential limitations for medical applications of the produced parts. Of the produced samples, 141 HV1 was the closest in hardness to that of human bone (40–79 HV1). Furthermore, a direct correlation between the various process parameters used for SLM and the measured hardness values was observed.

## 2. Materials and Methods

### 2.1. Elemental Powders

Pure Fe and Mn powders (Goodfellow Cambridge Ltd, Huntingdon, UK) were mechanically mixed to form a powder with 45.3 wt. % of Fe and 53.3 wt. % of Mn. To prepare the two powder mixtures (Fe-Mn and Fe-Mn-FeAg), a Shaker Mixer TURBULA® Type T2 C (Willy A. Bachofen AG, Basel, Switzerland) was used. The mixing process was performed for 1 h for each powder mixture before the experiments. A similar chemical composition measured with X-ray fluorescence spectroscopy (XRF) on a Thermo Scientific Niton XL3t GOLDD+ (Thermo Fisher Scientific, Massachusetts, MA, USA) was prepared with a 1.2 wt. % addition of pure silver (Ag) powder (Table 1). The metal powder properties were determined in accordance with ASTM B213 for the flowability (Hall flow meter), ASTM C1444 for the angle of repose, ASTM B212 for the apparent density and ASTM B527 for the tap density standards. The compression factor ($C_f$) was calculated with the equation

$$C_f = 100 \cdot \frac{(V_I - V_E)}{V_I} \tag{1}$$

and the Hauser ratio ($H$) was calculated with the equation

$$H = \frac{V_I}{V_E} \tag{2}$$

where $V_I$ is the initial volume and $V_E$ is the volume at the end of the tap density measurements.

**Table 1.** Chemical composition of powder mixtures. The remaining wt. % belongs to the oxygen.

| Samples | Sample Designation | Fe [wt. %] | Mn [wt. %] | Ag [wt. %] |
|---|---|---|---|---|
| **Powder mixtures** | Fe-Mn | 45.3 | 53.3 | / |
| | Fe-Mn-Ag | 44.7 | 53.3 | 1.2 |

### 2.2. SLM Production and Thermodynamic Investigation

Fe-Mn and Fe-Mn-Ag powder mixtures were analyzed with an STA449 C Jupiter (Erich NETZSCH GmbH & Co. Holding KG, Selb, Germany) and used for the SLM process. The DSC tests were performed with 10 K/min of heating and cooling (from 25 to 1550 °C and back to 25 °C). The isotherm was taken for 60 s. The SLM process parameters were changed so that the input energy density E was 146, 167 or 292 J/mm³. The printed model was a 10 mm cube. The simple line scanning strategy with a 54° rotation per layer was used for all the SLM-produced samples. The SLM-produced samples were created with two separate SLM processes by the Aconity MINI (Aconity 3D GmbH, Aachen, Germany), where exactly the same powder mixtures of Fe-Mn and Fe-Mn-Ag, the same working parameters and the same process conditions were used. In order to study the suitability and repeatability of the SLM process, we repeated the SLM process twice (A—first SLM process, B—repeated SLM process) with the same process parameters as well as with the same Fe-Mn powder mixtures. In one batch (A or B), eight cubes were printed. The measured oxygen values during the SLM process were at the initial support structure creation under 2000 ppm, and for the sample (cube) printing, the oxygen volume was 0 ppm. The differences between the chemical compositions of the SLM samples because of the inability to obtain an even distribution by mechanical mixing of the pure powders are presented in Table 1. The DSC analyses with the same conditions as for the powder mixtures were also performed for the SLM samples. For the sintering test, the powder mixtures were exposed to 850 °C for 1 h in ceramic pots.

### 2.3. Characterization of SLM-Produced Samples

The samples for the dilatometry were ground to a rectangular shape with dimensions of 3 mm × 3 mm × 8 mm. For the dilatometry with the Bähr DL 805A/D (TA Instruments

Inc., New Castle, DE, USA), the heating and cooling rates were 10 °C/s, with the isotherm at 1000 °C for 300 s. The tensile tests were made only on the set of samples with the lowest hardness in accordance with the DIN 50125 standard (quads, which were further shaped to the Metric ISO thread M6 testing probes). The samples were tested in an orientation perpendicular to the build direction. The Vickers hardness was measured with an Instron Tukon 2100B (Instron, Massachusetts, MA, USA) using a 10 kgf load. The Vickers hardness results are presented as the average of five measurements. The samples for observation on the macro- and micro-scales were ground and polished. Correlative microscopy between the light microscope (LM) Axiovision ZEISS AXIO Imager.Z2m (Carl Zeiss AG, Oberkochen, Germany) and the scanning electron microscope (SEM) with the ZEISS Crossbeam 550 FIB-SEM Gemini II (Carl Zeiss AG, Oberkochen, Germany) energy-dispersive spectroscopy (EDS) technique was performed with the EDAX Octane Elite EDS detector (AMETEK, Inc., Berwyn, IL, USA) with standard parameters (15 kV and 5 nA) and was used for the analysis of the sintered parts. For the crystallographic determination and observation of the $\varepsilon$-martensite and $\gamma$-austenite, the electron-backscatter diffraction (EBSD) technique with 19 kV and 5 nA (5–7 bands, with $4 \times 4$ binning) with the EDAX Hikari Super EBSD camera (AMETEK, Inc., Berwyn, IL, USA) was used. XRD analyses were performed with the PANalyitical 3040 (Malvern Panalyitical, Malvern, UK) X-ray diffractometer with Cu $K_\alpha$ radiation. The generated settings were 40 mA and 45 kV.

### 3. Results and Discussion

*3.1. Metal Powders and Chemical Composition of SLM-Produced Parts*

A high flowability, one of most important powder properties for SLM [24], could not be measured with a Hall flow meter for any of the powders. As a consequence, high angles of repose were measured. The angle of repose is, by definition, for a granular material, the steepest angle of descent (or dip relative to the horizontal plane) to which the material can be piled without slumping. Angles greater than 40° are observed for metal powders with poor flowability [25]. The mixture of Fe powder (48°) and Mn powder (55°) had a 49° angle of repose, which is closer to the lower value of both the pure powder components in the mixture. This is surprising and not in accordance with the mixture ratio where the higher percentage of Mn powder has a higher measured angle of repose. The powder mixture had a small difference between the apparent density (3.2 g/cm$^3$) and the tap density (3.8 g/cm$^3$) and a high compression factor (18.0), which is required for sintering as well as for additive manufacturing [26,27]. Despite all the negative characteristics of the powders and the powder mixtures, the SLM process was successful in producing the representative material. The chemical composition of the SLM-produced materials is presented in Table 2. For the deliberately porous material and the samples with defects that can accelerate the corrosion rate, powders with irregular shapes might be better as high-flowability metal powders in the form of spheres. The smaller addition of Ag powder to the powder mixture of Fe and Mn does not change the properties. The measured properties are presented in Table 3.

**Table 2.** Chemical composition of selective laser melting (SLM)-produced parts. SLM-produced samples. The number in the designation after the SLM-printed part´s chemical composition is the value of the energy density and the A or B mark means the first print or reprint. The remaining wt. % belongs to the oxygen (O).

| Samples | Sample Designation | Fe [wt. %] | Mn [wt. %] | Ag [wt. %] |
|---|---|---|---|---|
| | FeMn 146 A | 75.3 | 23.8 | / |
| | FeMn 146 B | 52.2 | 46.7 | / |
| | FeMn 167 A | 75.6 | 23.5 | / |
| | FeMn 167 B | 53.5 | 45.4 | / |
| **SLM-produced** | FeMnAg 292 A | 52.9 | 45.2 | 0.39 |
| | FeMnAg 292 B | 52.7 | 45.4 | 0.91 |
| | FeMnAg 167 A | 53.2 | 40.1 | 0.23 |
| | FeMnAg 167 B | 53.9 | 44.1 | 0.84 |

**Table 3.** Properties of metal powders.

| Powders | Angle of Repose [°] | Apparent Density [g/cm$^3$] | Tap Density [g/cm$^3$] | Compression Factor [/] | Hauser Ratio [/] |
|---|---|---|---|---|---|
| **Pure Fe** | 48 | 3.0 | 3.2 | 5.1 | 1.1 |
| **Pure Mn** | 55 | 2.3 | 3.1 | 24.2 | 1.3 |
| **Fe-Mn mix.** | 49 | 3.2 | 3.8 | 18.0 | 1.2 |

### 3.2. Differential Scanning Calorimetry

DSC records the heat flow at a given temperature (T). Each phase transformation is seen as a deviation from a baseline. The troughs represent endothermic reactions (melting), where energy is consumed. The peaks represent exothermic reactions (solidification), where energy is released [27]. The comparison of the two heating cycles (Figure 1a) shows the differences as well as the similarities. For the Fe-Mn powder mixture, the liquid temperature $T_L$ was determined to be 1326 ± 3 °C. A slightly lower $T_L$ of 1317 ± 3 °C was detected for the Fe-Mn-Ag powder mixture. The major difference appeared at 958 ± 3 °C, which is near to the Ag melting temperature and confirmed the presence of Ag in the powder mixture. For both curves, the path between 720 ± 3 and 948 ± 3 °C is the same and might be associated with the three phenomena. First, the susceptibility of the powder mixtures to sintering can be related to those changes in the heating curve. Secondly, the peak can belong to the melting of $MnO_2$, which is stable at room temperature and melts at 535 °C [28]. At elevated temperatures, the $MnO_2$ would dissociate into $Mn_2O_3$, $Mn_3O_4$ and $MnO$ [29]. The third option for the observed peaks could be the $\alpha_{Mn}$ transformation to $\beta_{Mn}$, which, under equilibrium conditions, is at approximately 700 °C. Figure 1b presents the cooling cycles of the Fe-Mn and Fe-Mn-Ag powder mixtures, where the two peaks are presented on both curves. The $T_{s1}$ was more similar (1499 ± 3 °C for the mixture without Ag and 1493 ± 3 °C for the mixture with Ag) than the $T_{s2}$. The first ($T_{s1}$) peak belongs to the start of the solidification of MnO. The presence of Mn oxides in the Mn powder was confirmed using SEM-EDS. The $T_{s2}$ for the γ-austenite solidification began at 1191 ± 3 °C and for Fe-Mn-Ag at a slightly higher temperature of 1221 ± 3 °C. The vaporization effect is, in accordance with the DSC results, more intensive with the Fe-Mn-Ag powder mixture, confirming the wide range of the vapor pressure for Ag.

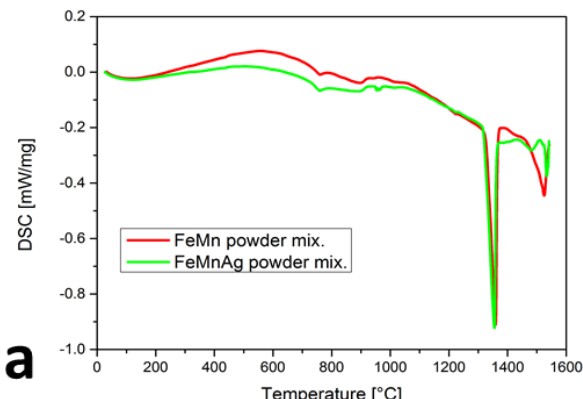
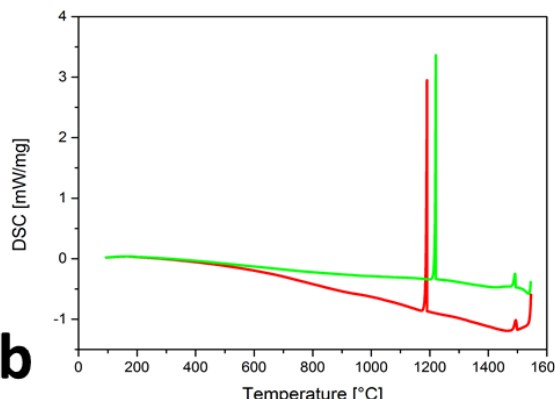

**Figure 1.** DSC results of Fe-M and Fe-Mn-Ag powder mixtures: (**a**) heating cycles, (**b**) cooling cycles.

Figure 2a shows the similarities of the chemical compositions between FeMn 146 A and FeMn 167 A with the lower Mn content as well as FeMn 146 B and FeMn 167 B with more Mn. For the first two (23.8 and 23.5 wt. % Mn), the $T_L$ is higher (approximately 1420 ± 3 °C) than for the other two (46.7 and 45.4 wt. % Mn), where $T_L$ is 1346 ± 3 and 1413 ± 3 °C. It is also evident that the ε-martensite appeared at 183 ± 3 °C for both curves, but only in alloys with a lower Mn content. The claims of Mujica et al. [30] that an increase in the

Mn will reduce the possibilities of ε-martensite's occurrence were proven. The reappeared changes in the temperature interval at approximately 700 to 1000 °C were observed. An even stronger detection appeared for the samples with Ag (Figure 2c), at the onset of the Ag melting and also in this temperature interval. As predicted previously with powder mixtures, the behavior of the curves in this temperature interval is associated with the combination of susceptibility to sintering, $MnO_2$ melting and Mn oxides transformation, as well as the $\alpha_{Mn}$-to-$\beta_{Mn}$ transformation. In Figure 2b,d, very narrow solidification intervals for the FeMn and FeMnAg alloys were found. The high vaporization of Mn with heating and under isothermal conditions completely changes the chemical composition of the analyzed alloys, which is evident from the $T_s$ temperatures. They were between $1202 \pm 3$ and $1391 \pm 3$ °C. Only for the sample FeMnAg 292 A, the second peak, at around $1330 \pm 3$ °C, appeared and probably belongs to the Mn oxides' transformations. The ε-martensite appeared only in the samples where the Mn content was low enough. The reason for the temperature differences in the ε-martensite's start could be the range of the chemical composition or the fact that the ε-martensite can occur athermically [31], which means that for the phase transformation, a change in the temperature is not required, but as a consequence of the phase transformation, a suppressed dislocation motion with stress delocalization is also present with the appearance of extensive phase boundaries.

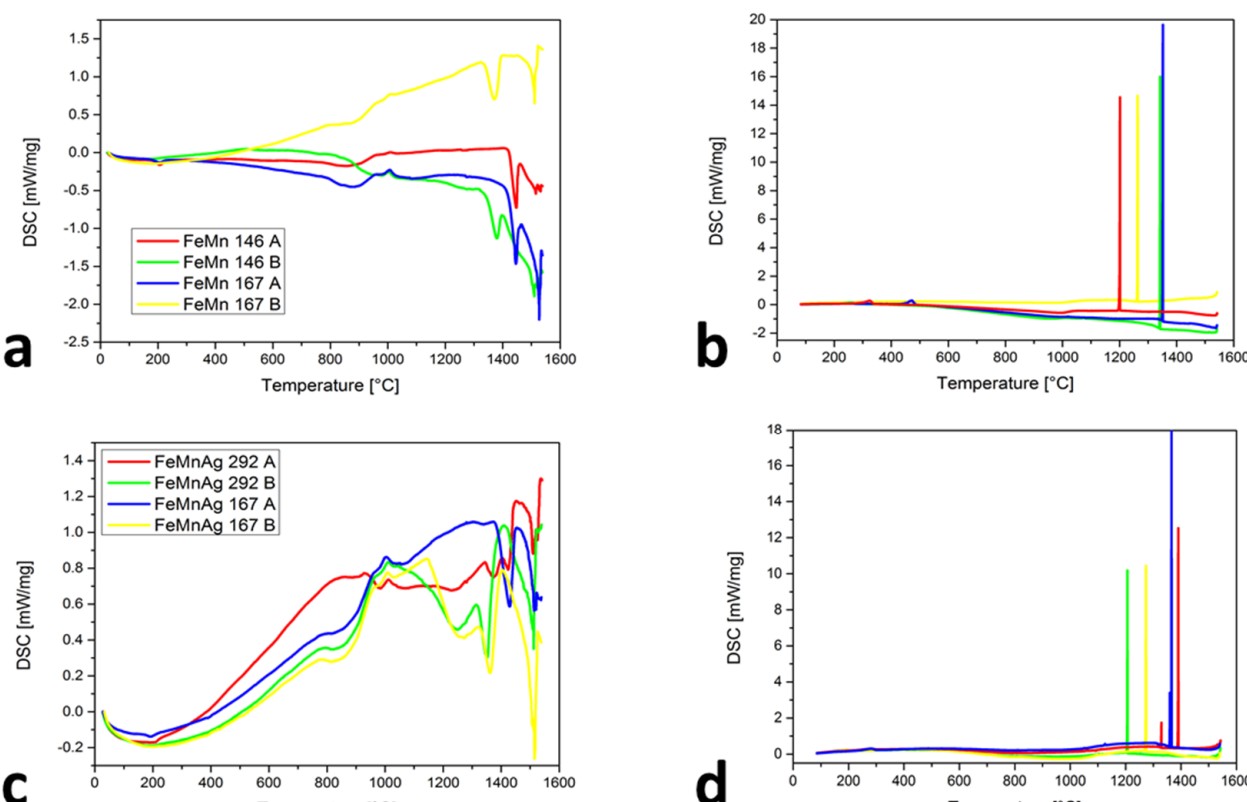

**Figure 2.** DSC results of SLM samples: (**a**) heating cycles for Fe-Mn samples, (**b**) cooling cycles for Fe-Mn samples, (**c**) heating cycles for Fe-Mn-Ag samples, (**d**) cooling cycles for Fe-Mn-Ag samples.

Problems with the vapor pressure differences and consequently the strong vaporization of Mn bring major challenges with respect to achieving the desired chemical composition. Especially for the SLM process, where the temperatures in the melt pool area are typically in the range of 1500 to 2500 °C [32] (but can also reach 3700 °C in the center of the laser spot [33]), the addition of Mn must be greater in the Fe-Mn powder mixture than the desired Mn content in the SLM samples. The undesirable chemical composition changes also arise from re-melting due to the SLM process, presented in the scheme (Figure 3a). The

marked positions show the different melted and re-melted spots during the SLM-produced layers and the behavior of the material in these spots during the heating (Figure 3b), the isotherm (Figure 3c) and the cooling (Figure 3d). With heating cycles, the $T_L$ value increases, which means an increasingly smaller Mn content is observed after each re-melting. The above is also confirmed with 60 s of the isotherm curves (that is approximately the same as the processing time for three layers) because the DSC values were constantly decreasing. The most representative are the cooling cycles, where the $T_s$ is much closer to the $T_s$ of pure Fe with each re-melting.

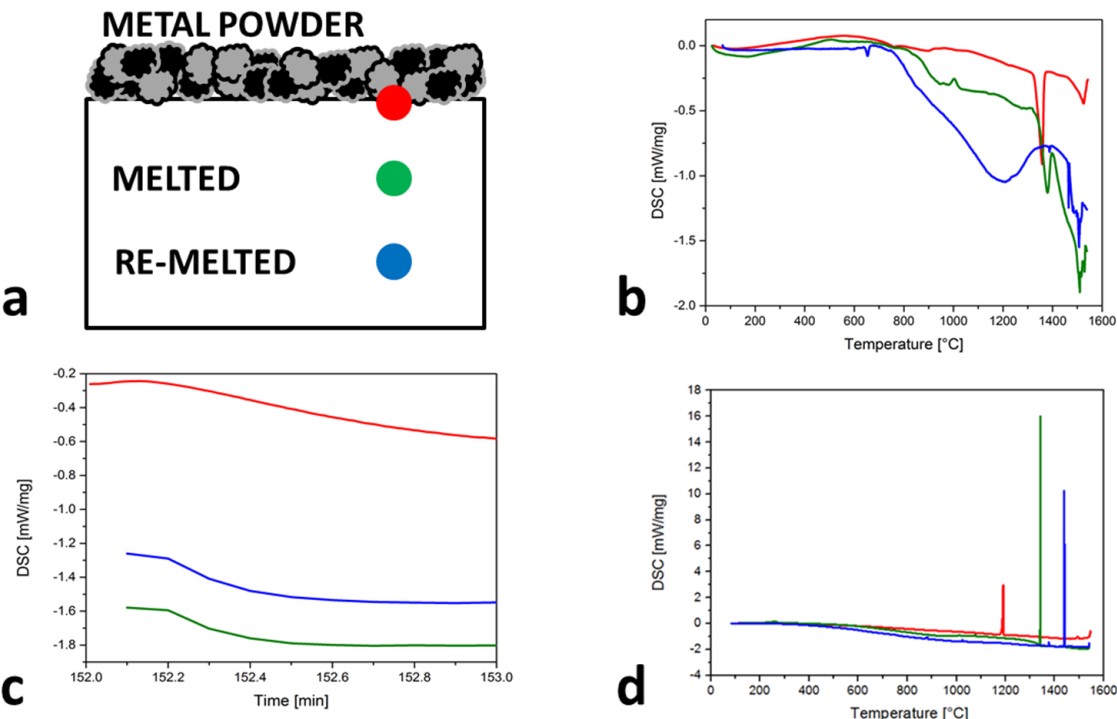

**Figure 3.** Solidified, melted and re−melted spots behavior of Mn content losses during the SLM process described by: (**a**) scheme with markers; (**b**) heating cycles; (**c**) isotherms; (**d**) cooling cycles.

Based on the strong vaporization and the consequently difficult-to-achieve desired chemical composition, it is undesirable to use mechanically mixed powders of Fe-Mn or Fe-Mn-Ag for the SLM process.

### 3.3. Sintering

The mechanically mixed Mn and Fe powders were sintered in air, as presented in Figure 4a (the positions of EDS and XRD analyses are marked). Upon exposure of the Fe-Mn or Fe-Mn-Ag powder mixture to 850 °C for 1 h (normal conditions for the major area of powder mixture in the working chamber during the SLM process), a highly compact material with three different layers was formed. The first layer (Figure 4b) was heavily oxidized and almost separated from the rest of the sample. This thin layer of oxidized Mn and Fe appeared all over the cross-section edge. Due to the sintered structure, the surface topography is rather rough, which enabled the precise EDS analysis, particularly for light elements (oxygen) [34]. However, the EDS maps adequately match the oxygen with other elements (Fe or Mn). The second layer is presented in Figure 4c,d. In this layer, in contrast to the first layer, the Mn is more oxidized than the Fe, which is reflected in the appearance of pure-Fe areas in the microstructure. The Ag powder does not melt or vaporize until 850 °C, as proven by the same shape of Ag powder particles in the powder mixture and in the sintered part. The SLM reused and sieved powder mixture can also contain solidified parts that occurred through the process and were mixed in the powder. During sintering,

these parts are often surrounded by Mn oxides. In Figure 4e, the most extensive areas with visible Fe and Mn particles are compressed and connected to the highly compact material. The mass of samples after sintering was increased due to the oxides' formation, which was also a frequent phenomenon in the corrosion tests [35]. The chemical composition of the parts after sintering had approximately 3 wt. % less Mn than the initial powder mixture. The vaporization problem related to the high vapor pressure of the Mn is again confirmed. Figure 4f presents XRD spectra of the sintered powder. The majority of phases are $\alpha$-Fe and $\alpha$-Mn. Some weak peaks in the spectra correlate to the $MnO_2$ phase and prove the presence of Mn oxides in the material.

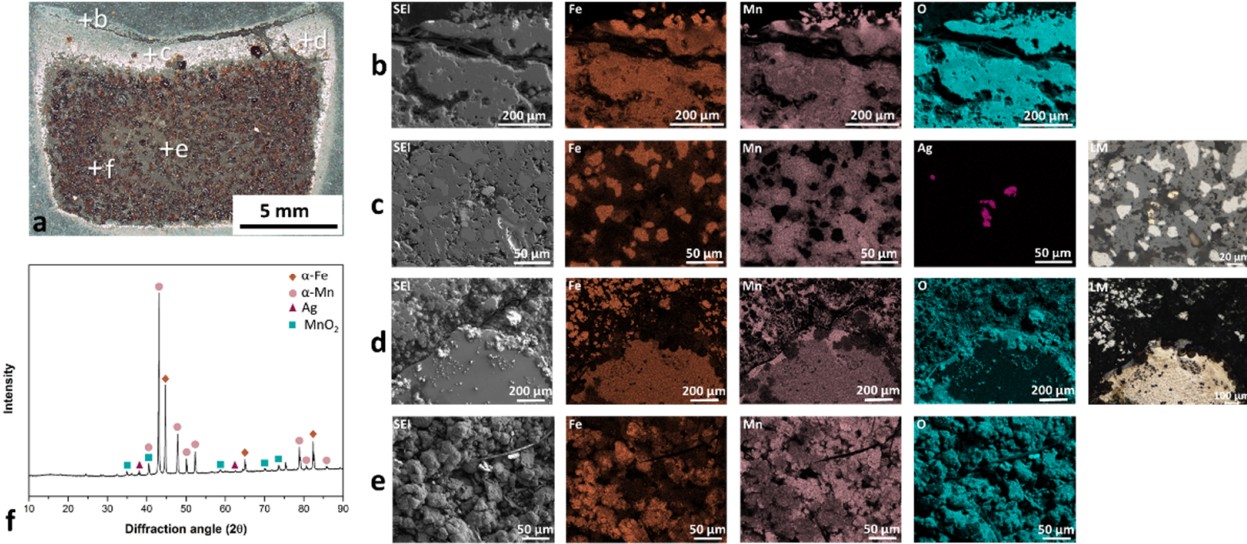

**Figure 4.** Sintering test for Fe-Mn-Ag powder mixture at 850 °C for 1 h: (**a**) macrograph of cross-section, (**b**) upper fully oxidized layer, which was partly resigned, (**c**) area with Mn oxide and completely pure Fe areas, (**d**) melted part as a result of the reused powder after the SLM process, surrounded with the sintered powder mixture, (**e**) in the mainly well-sintered area, (**f**) XRD spectra of sintered Fe-Mn-Ag material.

### 3.4. Dilatometry

The dilatometry curve (Figure 5a) shows that the $\varepsilon$-martensite transformation was present at low temperature (289 °C). The sintered layers and Mn oxides on the SLM samples most often occurred on the surfaces of the material. This was removed with grinding before the dilatometry tests. In that way, the Mn oxides' transformations would not be detected by the dilatometry curve. The inverse pole figure in the Z direction (IPF-Z) in Figure 5b presents the diverse orientation of the $\varepsilon$-martensite needles and the $\gamma$-austenite grains. The occurrence of the $\varepsilon$-martensite needles or plates, which intersect each other within the large $\gamma$-austenite grains, is described by Takaki et al. [36]. In Figure 5c, the ratio between the $\gamma$-austenite with the face-centered cubic (FCC) structure and the $\varepsilon$-martensite with the hexagonal close-packed (HCP) structure is presented.

### 3.5. Hardness

Changing the Mn and Ag contents in the chemical composition, on the one hand, and the SLM parameters, on the other, results in materials between 141 $\pm$ 4 and 276 $\pm$ 2 HV (Table 4). The lower Mn content in the alloys causes an easier transformation of $\gamma$-austenite to $\varepsilon$-martensite, which is harder. In a comparison with the hardness of human bone, the lowest hardness was detected for sample FeMn 167 B, which is twice as hard as bone [16]. For the same material, the UTS is 553 $\pm$ 10 MPa, the YT is 331 $\pm$ 8 MPa and the A is 3.8 $\pm$ 1%, which is similar to the A value for human bone [8].

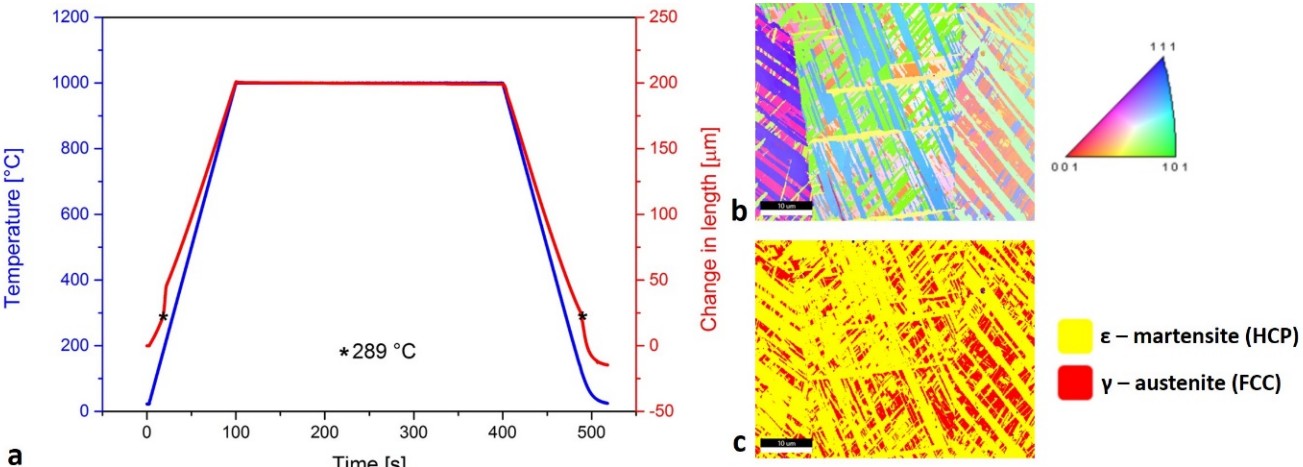

**Figure 5.** Presentation of ε-martensite: (**a**) dilatometry curve with marked same temperature of transformation during heating and cooling; (**b**) inverse pole figure in Z direction (IPF-Z) where the diversity of ε-martensite and γ-austenite orientation is presented; (**c**) ratio between γ-austenite and ε-martensite with specific type of needle formation.

**Table 4.** Hardness values for SLM samples.

| Sample Designation | Hardness [HV1] |
|---|---|
| FeMn 146 A | 211 ± 5 |
| FeMn 146 B | 145 ± 3 |
| FeMn 167 A | 276 ± 2 |
| FeMn 167 B | 141 ± 4 |
| FeMnAg 292 A | 264 ± 6 |
| FeMnAg 292 B | 143 ± 7 |
| FeMnAg 167 A | 246 ± 5 |
| FeMnAg 167 B | 154 ± 4 |

## 4. Conclusions

The thermodynamic behavior and the suitability of the SLM process for mechanically mixed Fe-Mn and Fe-Mn-Ag powders were investigated. The differences between the chemical compositions of the initial powder and the SLM sample were explained by the strong tendency for Mn to vaporize during the DSC analyses. The response of both powder mixtures in the SLM working chamber was reconstructed for 1 h at 850 °C. The SLM samples were also used for the dilatometry to observe the transformation from the γ-austenite to the ε-martensite. Based on the microstructural results and the mechanical properties, the following conclusions can be drawn:

- It is difficult to manufacture an SLM product from powder mixtures of elements, particularly with the presence of Mn, a high-vapor pressure element, and especially with Mn oxides, which are present in Mn powder. Mn oxides have several transformations in the temperature interval from 700 to 1000 °C, confirmed by the DSC.
- On the DSC cooling curves, the vapor of Mn and, therefore, the changes in the chemical composition with each re-melting were observed as a shift in the solidification start temperature.
- One hour of exposure of the powder mixtures to 850 °C resulted in strong, compact and well-sintered parts. At the same time, the Mn content in comparison to the powder mixtures decreased by 3 wt. %.
- Complex intersections of ε-martensite needles or plates within large γ-austenite grains were observed. This is the microstructure shape type that occurs at lower temperatures (298 °C) and cannot be avoided during the SLM process. Furthermore, these products are not ideal for medical applications.

- The lower Mn content in the SLM samples increased the probability of the ε-martensite transformation, which was confirmed by the hardness measurements. The presence of ε-martensite in the microstructure can almost double the hardness of the material.

**Author Contributions:** Conceptualization, J.K. and J.M.; methodology, J.K. and J.M.; software, J.K.; validation, J.M.; formal analysis, J.K. and I.P.; investigation, J.K. and I.P.; resources, M.G. and J.M.; writing—original draft preparation, J.K.; writing—review and editing, I.P., M.G. and J.M.; visualization, J.K. and M.G.; supervision, I.P., M.G. and J.M.; funding acquisition, M.G. All authors have read and agreed to the published version of the manuscript.

**Funding:** The authors acknowledge the financial support from the Slovenian Research Agency, research core funding No. J2-1729 and P2-0132.

**Conflicts of Interest:** The authors declare no conflict of interest.

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
