# Peer review of "Thermodynamic Behavior of Fe-Mn and Fe-Mn-Ag Powder Mixtures during Selective Laser Melting"

_metals, doi:10.3390/met11020234_

Round 1

Reviewer 1 Report

Additive manufacturing from elementary powders in the SLM process is very promising due to the elimination of the compiled time-consuming and energy-consuming process of producing alloyed powders, so the article “Thermodynamic Behavior of FeMn and FeMnAg Powder Mixtures during Selective Laser Melting” is an interesting publication with a significant degree of novelty. The experiments were carried out logically, and the results are generally presented correctly, however, it requires the authors to make some contribution to the arrangement of the layout in order to make the article more transparent and accessible to the reader. Authors should correct and/or add the following areas prior to publication:

  1. Whole article: avoid jargon: “all this”, “very complex” - with internal channels? With open porosity? Cellular solids ?, “good microstructures”, “good biocompatible” “some differences” “may similarities” etc. Be precise and write what you mean and what is stated in the literature.
  2. Abstract - properties of initial metal powders impact the final properties of any powder metallurgy technique. Please avoid this kind of unprecise statements. You should not finish the abstract with a statement with just elongation because you have measured also mechanical properties.
  3. Introduction: There is a lack of information in the introduction about the state of the art on the influence of flowability, processing parameters, and scanning strategies on properties of SLM processed parts. There is only described well why you have chosen FeMn alloy. The novelty of the work is not stated clearly. It should be described precise.
  4. The materials and methods section is very disordered. Divide it into sections: f.e. 1. Elemental powders description + describe here how you measured flowability 2. Samples design and SLM fabrication parameters - A & B option are described unclear. 3. Bulk samples characterization methods. Properties of SLM produced parts should be in the results section!
  1. Results:
  • Flowability results are described unclear. Describe here your connection to reference - 21 and 22. Maybe that should be in the introduction? Describe what is the angle of repose. The graph would be beneficial. How you calculated values in Table 2? This should be in materials and methods.
  • Add% of theoretical density for FeMn samples. Why just one sample was measured when you have received changes of hardness after remelting?
  • Statement "Melt-pool area are always between 1500-2500" is not true. They are typically in the range but on an open parameter system you can change its value, it would be also dependent on melted material
  • Describe laser scanning strategy and remelting scanning strategy in materials and methods and in results sections.
  • You should remove oxygen results from EDS maps. You can see here just the topography of the surface with oxygen contamination.
  • The type of oxides was not measured correctly. You should use XRD diffraction. Different phases could be also shown using optical microscopy.
  • Why you have received changes in sample hardness after second processing? Why hardness is decreasing after the second melting? Why higher energy density was used for samples with Ag?
  • Why dilatometry is shown with EBSD? Why EBSD is not described in materials and methods?

Author Response

Dear editors and reviewers:

Thank you very much for your careful review and constructive suggestions with regard to our manuscript entitles “Thermodynamic Behaviour of Fe-Mn and Fe-Mn-Ag Powder Mixtures during Selective Laser Melting”. We have considered the comments carefully, and then revised and improved the manuscript according to these comments. The main corrections to the paper (marked with yellow in the manuscript) and the responses to the reviewers’ comments are below:

  1. Whole article: avoid jargon: “all this”, “very complex” - with internal channels? With open porosity? Cellular solids ?, “good microstructures”, “good biocompatible” “some differences” “may similarities” etc. Be precise and write what you mean and what is stated in the literature.

  • Thank you for your comments. In accordance with your suggestions, the manuscript was re-written. We have avoided the use of imprecise expressions.
  • The whole article was corrected by a proof-reader, Dr Paul John McGuiness.
  • The proofreading certificate is attached at the end of this document.

  1. Abstract - properties of initial metal powders impact the final properties of any powder metallurgy technique. Please avoid this kind of unprecise statements. You should not finish the abstract with a statement with just elongation because you have measured also mechanical properties.

  • Thank you for your comment. We corrected the imprecise part of the statement. Additive manufacturing is a form of powder metallurgy, which means the properties of the initial metal powders (chemical composition, powder morphology and size) impact on the final properties of the resulting parts.
  • In medical applications, especially when making comparisons with human bone, the metals or alloys have a higher UTS and YS. Therefore, the approximation with the elongation is much more important. The sentence in the abstract was re-written accordingly.

  1. Introduction: There is a lack of information in the introduction about the state of the art on the influence of flowability, processing parameters, and scanning strategies on properties of SLM processed parts. There is only described well why you have chosen FeMn alloy. The novelty of the work is not stated clearly. It should be described precise.

  • Thank you for your comment. In the end of introduction we clearly stated the novelty and purpose of our work.
  • We added to the manuscript: “The focus of our study was to find out whether it is possible to produce FeMn parts with appropriate properties using elementary powder mixtures by SLM technique. Despite the fact from Fe-Ag phase diagram that Fe and Ag are not mixing in liquid neither solid state was our focus to prepare FeMnAg alloy by rapid solidification process using SLM.” Our aim was not to study the influence of the process parameters or the scanning strategy, but to focus on an investigation of the possibilities and influences of the elements-based powder mixture and the differences in the properties of the produced parts, as in the manufacture of alloys and an understanding of the more complex powder without flowabillity.
  • Our investigation was novel in terms of a thermodynamic understanding of the high-vapour-pressure material (Mn) and its alloying impact, and not on the SLM’s process parameters (scanning strategy).

  1. The materials and methods section is very disordered. Divide it into sections: f.e. 1. Elemental powders description + describe here how you measured flowability 2. Samples design and SLM fabrication parameters - A & B option are described unclear. 3. Bulk samples characterization methods. Properties of SLM produced parts should be in the results section!
  • Thank you for this suggestion. The section was ordered in accordance with your suggestions. Also, the A and B options were described in more detail.
  • The SLM-produced samples were created with two separate SLM processes, where exactly the same powder mixtures of the Fe-Mn and of the Fe-Mn-Ag, the same working parameters and process conditions were used. In order to study suitability and repeatability of the SLM process we repeated SLM process twice (A – first SLM process, B – repeated SLM process) with same process parameters as well as with the same Fe-Mn powder mixtures.
  • The process parameters are described in terms of the input energy and the scanning strategy.
  • The SLM properties were moved to the results and discussion section, as suggested.

  1. Results:

Flowability results are described unclear. Describe here your connection to reference - 21 and 22. Maybe that should be in the introduction? Describe what is the angle of repose. The graph would be beneficial. How you calculated values in Table 2? This should be in materials and methods.

  • The flowabillity of the used powder mixtures cannot be measured (because there is no flowabillity), which is also clear from the high values for the angle of repose. It is unusual to involve powders with such a low flowability for SLM process, but the economic and novelty reasons are sufficient justification to make these experiments. The powders of Fe and Mn are usable for SLM, but as stated in conclusions it is not an advisable strategy.
  • The flowabillity measurements are in the results and discussion section, because the lack of flowabillity of the powders was discovered after the experimental work.
  • The angle of repose is, by definition, for a granular material the steepest angle of descent (or dip relative to the horizontal plane) to which the material can be piled without slumping. This fact has been added to the text.
  • We added the equations for the calculations from Table 2.

Add% of theoretical density for FeMn samples. Why just one sample was measured when you have received changes of hardness after remelting?

  • The theoretical density of the FeMn (45.4 wt. % Mn) sample is 7.69 g/cm3, without any oxygen. Because we did not measure the porosity of the SLM-produced samples, we did not mention any theoretical density.  
  • The samples produced for the tensile test, especially with powders of poor quality, were difficult and dangerous for the SLM machine. Changing the parameters did not help, but made the situation more complex and difficult. In most cases the SLM processes of manufacturing probes for the tensile tests were not completed, because the powders, parameters or shapes were not appropriate. For our investigation the properties of the human bone were the orientation, and in that way we determined the UTS, YS and elongation just for the lowest hardness values of the FeMn material.

Statement "Melt-pool area are always between 1500-2500" is not true. They are typically in the range but on an open parameter system you can change its value, it would be also dependent on melted material

  • Thank you for raising this point. We totally agree with you. The sentence was corrected. We referred to the sentence and mentioned in the discussion why these high values have a negative impact on our powder mixtures.

Describe laser scanning strategy and remelting scanning strategy in materials and methods and in results sections.

  • Thank you for your comment. We added a description of laser-scanning strategy in the text: “The simple line scanning strategy with 54 ° rotation per each layer was used for all SLM produced samples”.
  • During the SLM process, each laser scan influences the already-solidified material under the scanning area (see figure below), where the re-melting of the material occurs. There was no particular re-melting scanning strategy. In our investigation the re-melting process was simulated with repeated DSC experiments.

You should remove oxygen results from EDS maps. You can see here just the topography of the surface with oxygen contamination.

  • With respect to your comment, we still think that the EDS oxygen map shows the oxygen distribution and not only the surface contamination. It is evident from the EDS maps that the oxygen correlates with the manganese.

The type of oxides was not measured correctly. You should use XRD diffraction. Different phases could be also shown using optical microscopy.

  • Thank you for your comment. We are aware that EDS is not the proper technique for oxide-type determination. We only confirmed the presence of oxygen in our samples.
  • As presented in the micrographs that were well etched, the εM phase cannot be seen with a light microscope. To prove its existence and observe the εM, EBSD was the best solution.

Why you have received changes in sample hardness after second processing? Why hardness is decreasing after the second melting? Why higher energy density was used for samples with Ag?

  • Thank you also for this comment. There was no second processing. As previously explained, the re-melting is an unintentional and unavoidable phenomenon. The hardness decreased with the higher percentage of Mn because the increased amount of Mn in the FeMn alloys slows down the εM The εM is harder than pearlite, ferrite or austenite. This is now mentioned in the manuscript.
  • In two particular cases, exactly the same energy density was used for the FeMn and FeMnAg. However, when higher energy density for Fe-Mn-Ag powder mixtures was used better mechanical properties (nearer to the human bone) were achieved. In the article we present only the best results for the FeMn and FeMnAg material. We proved that for the FeMnAg alloys, a higher energy density results in better properties.

Why dilatometry is shown with EBSD? Why EBSD is not described in materials and methods?

  • Thank you for this comment. The connection between the dilatometry and the EBSD is εM. With the combination of both analysing techniques we presented the temperature of the εM formation and the εM That made the progress in understanding of εM transformation, which is undesirable for SLM-produced FeMn alloys.
  • The EBSD is described in the recommended section. For the crystallographic determination and observation of the ε-martensite and γ-austenite, the EBSD technique was used.     

We also changed the FeMn in to the Fe-Mn in the whole manuscript, what is technically more correct.

We hope that our explanations and corrections will meet with your approval.

Yours sincerely,

Jakob Kraner

Institute of Metals and Technology

Reviewer 2 Report

The paper lacks details in the following areas:

Need to compare results with phase diagrams

Explain how the powders were mechanically mixed

Manufacturer of Vickers hardness

Table 1 - weight percents do not add up to 100 - what are the other constituents

How accurate is the DSC - +/- 10 Deg C?

Explain on page 5 - can occur athermically

Figure 3 - Explain what the peaks and valleys mean

Need standard deviation on the hardness measurements

UTS data needs to have standard deviation

Is there a time element in SLM that is not captured in the DSC data?

How many hardness measurement were made?

The conclusions are weak - and obvious

They need to be rewritten - The analyzed powder mixtures are challenging for SLM - what does that mean?  FeMn powders are difficult to process with SLM?

If you reduce the Mn content, the melting temperature should move closer to Fe - 

Why are complex interacts that occur at low temperatures of interest?

Author Response

Dear editors and reviewers:

Thank you very much for your careful review and constructive suggestions with regard to our manuscript entitled “Thermodynamic Behaviour of Fe-Mn and Fe-Mn-Ag Powder Mixtures during Selective Laser Melting”. We have studied the comments and revised and improved the manuscript in accordance with the comments. The corrections in the manuscript and the responses to the comments are as follows:

The paper lacks details in the following areas:

  1. Need to compare results with phase diagrams
  • The binary as well the ternary phase diagrams were calculated for combinations of Fe, Mn, Ag and O. The equilibrium and non-equilibrium solidifications were calculated with Thermocalc. Some examples are presented in the following figures.

  • There are a lot of reasons why the comparison does not contribute to the results. With the SLM process and the DSC experiments, the Mn uncontrolled vaporized and that way the chemical composition, which is crucial at thermodynamic calculations, is constantly changing. The mechanically mixed powders are not homogeneous and therefore the phase diagram calculations is challenging. By the non-equilibrium calculation (Shail calculation) for all chemical compositions of powders and SLM produced parts only γFe (austenite) is present in microstructure. As presented in article with larger amount of εM (epsilon martensite) also these calculations are not comparable with the experiment results.    

  1. Explain how the powders were mechanically mixed
  • Thank you for this correction. We added a description in the part about the materials and methods.
  • For mixing the two powder mixtures (FeMn P and FeMnAg P) a Shaker Mixer TURBULA® Type T2 C (Willy A. Bachofen AG, Basel, Switzerland) was used. The mixing process was performed for 1 h for each powder mixture before using for different experiments (SLM, sintering, properties measurements).

  1. Manufacturer of Vickers hardness
  • Thank you for your comment. The manufacturer is added.
  • The Vickers hardness was measured with Instron Tukon 2100B (Instron, Massachusetts, USA) using a 10 kgf load.

  1. Table 1 - weight percents do not add up to 100 - what are the other constituents
  • Thank you for this comment. In the Table 1 caption the justification for the elements was added.
  • The remaining wt. % belongs to the oxygen, which is in the form of oxides. The oxides are undesirable, because we don’t want them in SLM manufactured parts.
  • The XRF measurements are accurate for Fe, Mn, Ag. The balance is O and can be calculated.

  1. How accurate is the DSC - +/- 10 Deg C?
  • The DSC with an S-type thermocouple is accurate to ±25 % of the measured temperature. In the worst case that would be ± 3 °C. The measurements uncertainty was added accordingly.

  1. Explain on page 5 - can occur athermically
  • Thank you also for this important note. The suggested explanation is added.
  • “…which means that for the phase transformation a change in the temperature is not required, but as a consequence of phase transformation a suppressed dislocation motion with stress delocalization is also present with the appearance of extensive phase boundaries.”

  1. Figure 3 - Explain what the peaks and valleys mean
  • The base curve by the DSC measurements is recorded heat flow, which is in accordance to the temperature (T). That way is each phase transformation recognized as deviation from base curve. The valleys present endothermal reactions (melting), where the energy is consumed. Opposite, the peaks presents exothermal reactions (solidification), where the energy is released.
  • The statement is also added to the manuscript.

  1. Need standard deviation on the hardness measurements
  • Thank you for your comment. Standard deviations of the hardness measurements were added.

  1. UTS data needs to have standard deviation
  • Thank you for your comment. Standard deviations for the UTS data were added.

  1. Is there a time element in SLM that is not captured in the DSC data?
  • Thank you for your comment. There is no time element in SLM that should be captured in the DSC data. The connection that is mentioned in the manuscript between DSC and SLM is that for one layer (single produced sample), the laser scan takes 20 s. So, if we made the DSC experiment with an isotherm of 60 s, this means that we simulate the process where the liquid is at some point 90 μm deep (between one and another layer is 30 μm) in produced part.

  1. How many hardness measurement were made?
  • Thank you for this question. An explanation with five measurements of the Vickers hardness was added in the part about materials and methods.
  • The Vickers hardness results are presented as an average of five measurements, which were performed for all the tested samples.

  1. The conclusions are weak - and obvious

They need to be rewritten - The analyzed powder mixtures are challenging for SLM - what does that mean?  FeMn powders are difficult to process with SLM?

If you reduce the Mn content, the melting temperature should move closer to Fe -

Why are complex interacts that occur at low temperatures of interest?

  • Thank you for your very valuable comments. The conclusions were re-written accordingly.

We also changed the FeMn in to the Fe-Mn in the whole manuscript, what is technically more correct.

We hope that the corrections will now meet with your approval.

Yours sincerely,

Jakob Kraner

Institute of Metals and Technology

Round 2

Reviewer 1 Report

Thank you for your answers and for improving the manuscript.

Before publication there still should be added/modified a few aspects:

  1. Add missing data in materials and methods (SLM machine information + how many samples were printed in 1 batch + oxygen volume during the process, EDS/SLM and EBSD parameters should be added.
  2. EDS measurements should be made ONE MORE TIME on the flat sample (https://onlinelibrary.wiley.com/doi/pdf/10.1002/sca.4950090205). The shadowing effect in figure 4 makes these results completely unacceptable especially in terms of oxygen validation. However, I would recommend adding XRD/XPS, etc. results to show oxides types.
  3.  Adding in the introduction the state of the art for the SLM printing from various elemental powders would be beneficial.

Author Response

Dear editors and reviewers:

Thank you very much for your careful review and constructive suggestions with regard to our manuscript entitles Thermodynamic Behaviour of Fe-Mn and Fe-Mn-Ag Powder Mixtures during Selective Laser Melting”. We have considered the comments carefully, and then revised and improved the manuscript according to these comments. The main corrections to the paper (marked with yellow in the manuscript) and the responses to the reviewers’ comments are below:

  1. Add missing data in materials and methods (SLM machine information + how many samples were printed in 1 batch + oxygen volume during the process, EDS/SLM and EBSD parameters should be added.
  • Thank you for your comment. We add missing data in materials and methods according to your suggestions.
  1. EDS measurements should be made ONE MORE TIME on the flat sample (https://onlinelibrary.wiley.com/doi/pdf/10.1002/sca.4950090205). The shadowing effect in figure 4 makes these results completely unacceptable especially in terms of oxygen validation. However, I would recommend adding XRD/XPS, etc. results to show oxides types.
  • Thank you for your comment. We agree that quantitative EDS analyses can only be performed on flat surfaces. It is not possible to metallographically prepare flat surface of sintered material. However, in our study qualitative results were required to see the match of oxygen with other present elements (Fe and Mn) what is clearly seen from EDS mapping. To confirm the oxide types, XRD analysed was performed as suggested. The XRD spectre was added and discussed in the manuscript.  
  1. Adding in the introduction the state of the art for the SLM printing from various elemental powders would be beneficial.
  • Thank you for your comment. We add the state of the art for the SLM printing from various elemental powders in the introduction accordingly.

We hope that our explanations and corrections will meet with your approval.

Yours sincerely,

Jakob Kraner

Institute of Metals and Technology
